# Error Analysis of Fitted Q-iteration with ReLU-activated Deep Neural Networks

**Lican Kang**
School of Mathematics and Statistics
Wuhan University
`kanglican@whu.edu.cn`

**Han Yuan**
Duke-NUS Medical School
National University of Singapore
`yuan.han@u.duke.nus.edu`

**Chang Zhu**
Tongji Hospital
Huazhong University of science and technology
`changzhu@hust.edu.cn`

## Abstract

Deep reinforcement learning (RL) has grown rapidly with the development of backbone feedforward neural networks (FNNs). However, there remains a theoretical gap when researchers conduct error analysis of the FNNs-based RL process. In this work, we provide an error analysis for deep-fitted $Q$-iteration applying ReLU-activated FNNs for value function approximation.

## 1 Introduction

Reinforcement learning (RL) has successfully trained sequential decision-making models over the last decade (Silver et al., 2016; Chen et al., 2020; Cao et al., 2022). Unlike conventional supervised training with explicit targets, RL aims to generate an agent maximizing the expected future return through implementing actions, interacting with the environment, and obtaining rewards. A well-behaved agent is to train a value function that maximizes the final reward. Such processes can be mathematically modeled as Markov Decision Processes (MDPs) defined in Appendix A.1. Deep RL replaced prior-defined value function with FNNs (Henderson et al., 2018). As a representative value-based algorithm in deep RL, deep-fitted Q-iteration (DFQI) takes transition data as its input and approximates the target value function using FNNs (Ernst et al., 2005). The statistical properties of traditional fitted Q-iteration (FQI) with function approximation are well-studied (Murphy, 2005). For DFQI, Fan et al. (2020) tried to provide a theoretical analysis.

Our work further complements Fan et al. (2020) in the following aspects: (1) Our error bound depends on the ambient dimension $d$ explicitly and polynomially (not implicitly and exponentially); (2) We introduce a weaker $\alpha$-mixing condition (Modha & Masry, 1996; Hang & Steinwart, 2014) than $\beta$-mixing in Antos et al. (2007) to characterize the dependency of MDPs (not assume that the batch data are independently and identically distributed ignoring the temporal dependency of MDPs); (3) We assume that the optimal action-value function $Q^*$ is a Hölder continuous function (without a composition form of certain functions).

## 2 Method and Results

Through the tools of error propagation (Proposition C.1), statistical error analysis (Theorem C.1), and deep approximation error analysis (Theorem C.2), a non-asymptotic error bound has been established between the estimated action-value function corresponding to the estimated greedy policy and the optimal $Q^*$ by controlling the statistical and approximation errors on MDPs assumed to be $\alpha$-mixing. Then we derive the generalization bound in terms of the ReLU-activated FNNs and $\alpha$-mixing data in the context of RL. Finally, we demonstrate that the error bound depends on the sample size, the ambient dimension (polynomially), the width and depth of the neural network, and the number of training iterations, and thus provide a powerful tool in hyper-parameters setting.

As the optimal Bellman operator $\mathcal{T}^*$ in (5) is a $\zeta$-contraction, we can use fixed point iteration at the population level to approximate the optimal action-value function $Q^*$ in (4). Specifically, when $\mathcal{R}(\cdot|x, a)$ and $P(\cdot|x, a)$ are known, the following iteration (1) can approximate $Q^*$ well if $J$ is large enough, i.e.,

$$Q_0 \to Q_1 = \mathcal{T}^* Q_0 \to Q_2 = \mathcal{T}^* Q_1 \to \ldots \to Q_J = \mathcal{T}^* Q_{J-1}. \tag{1}$$

Here, $Q_{j-1}$ and $Q_j$, $j = 1, \ldots, J$, satisfy

$$Q_j \in \arg\min_Q \mathcal{L}(Q) = \mathbb{E}|Q(X, A) - Y|^2. \tag{2}$$

In (2), $\mu$ represents the distribution of the state-action pair $(X, A)$, and $Y = R + \zeta \max_{a' \in \mathcal{A}} Q_{j-1}(X', a')$. However, in practice, we only have access to the batch data $\{Z_i\}_{i=1}^n = \{X_i, A_i, R_i, X_i'\}_{i=1}^n$. Therefore, DFQI uses an estimator $\widehat{Q}_j$ in $\mathcal{F}$ to mimic the fixed point iteration in (1). The estimator $\widehat{Q}_j$ is obtained by solving the regression problem

$$\widehat{Q}_j \in \arg\min_{Q \in \mathcal{F}} \widehat{\mathcal{L}}(Q) = \frac{1}{n} \sum_{i=1}^n \left( Q(X_i, A_i) - Y_i \right)^2, \tag{3}$$

where $\mathcal{F}$ represents the ReLU-activated FNNs, and $\widehat{Q}_0 \in \mathcal{F}$ is an initial guess, and $Y_i = R_i + \zeta \max_{a' \in \mathcal{A}} \widehat{Q}_{j-1}(X_{i+1}, a')$. It is important to note that the empirical loss in (3) is an unbiased estimation of the population loss in (2), i.e., $\mathbb{E}[\widehat{\mathcal{L}}(Q)] = \mathcal{L}(Q)$, $\forall Q \in \mathcal{F}$. Details of the DFQI algorithm are provided in Algorithm 1.

We introduce the definition of $\alpha$-mixing for describing the dependence of a stochastic process.

**Definition 2.1.** ($\alpha$-**mixing**) *Let $\{U_t\}_{t \geq 1}$ be a stochastic process. Denote by $U^{1:n}$ the collection $(U_1, \ldots, U_n)$, where we can allow $n = \infty$. Let $\sigma\left(U^{i:j}\right)$ denote the $\sigma$-algebra generated by $U^{i:j}(i \leq j)$. The $m$-th $\alpha$-mixing coefficient of $\{U_t\}_{t \geq 1}$, $\alpha_m$, is defined by*

$$\alpha_m = \sup_{t \geq 1} \sup_{A \in \sigma(U^{1:t}), B \in \sigma(U^{t+m:\infty})} |\mathbb{P}(AB) - \mathbb{P}(A)\mathbb{P}(B)|.$$

*$\{U_t\}_{t \geq 1}$ is said to be $\alpha$-mixing if $\alpha_m \to 0$ as $m \to \infty$. We say that a $\alpha$-mixing process is exponential if there exists parameters $\bar{\alpha}, a, \eta > 0$ such that $\alpha_m \leq \bar{\alpha} \exp\left(-am^\eta\right)$ holds for all $m \geq 0$.*

Now, we give the main result in this paper, a non-asymptotic error bound of DFQI.

**Theorem 2.1.** *Suppose that $\{\mathcal{T}^* \widehat{Q}_{j-1}\}_{j=1}^J \in \mathcal{H}^\gamma$ in Definition C.2 with $\gamma = s + r, s \in \mathbb{N}_0$ and $r \in (0, 1]$, $\{Z_i\}_{i=1}^n$ is strictly exponentially $\alpha$-mixing in Definition 2.1, and the probability distribution $\mu$ of $(X, A)$ is absolutely continuous with respect to Lebesgue measure. Then, for the ReLU-activated FNNs $\mathcal{F}$ with the width $\mathcal{W} = \mathcal{O}\left((n^{\frac{\eta}{1+\eta}})^{\frac{d}{4(d+4\gamma)}} \log n\right)$ and depth $\mathcal{D} = \mathcal{O}\left((n^{\frac{\eta}{1+\eta}})^{\frac{d}{4(d+4\gamma)}} \log n\right)$, we have*

$$\mathbb{E}\left[\|Q^* - Q^{\pi_J}\|_{L_1(\nu)}\right] \leq \frac{2\sqrt{C}C_{\nu,\mu}\zeta}{(1-\zeta)^2} \cdot \left[d^{s+(\gamma \vee 1)/2}(n^{\frac{\eta}{1+\eta}})^{\frac{-\gamma}{d+4\gamma}}(\log n)^{3/2}\right] + \frac{4\zeta^{J+1}}{(1-\zeta)^2}R_{\max},$$

*where $C$ is a constant depending on $s, B, \mathcal{B}, R_{\max}, \eta, a, \bar{\alpha}$.*

The mild completeness condition, denoted by $\{\mathcal{T}^* \widehat{Q}_{j-1}\}_{j=1}^J \in \mathcal{H}^\gamma$, used in Theorem 2.1 is satisfied when the MDP underlying the problem satisfies certain smoothness conditions, as pointed out in Fan et al. (2020). Moreover, it is also noted that a completeness condition is necessary, as highlighted in Chen & Jiang (2019). Theorem 2.1 provides a non-asymptotic error bound, that is $\mathcal{O}\left(n^{\frac{-\gamma\eta}{(1+\eta)(d+4\gamma)}}\right)$ when $J$ is large enough. This bound improves upon the result in Fan et al. (2020) by taking into account the temporal dependence of data and by reducing the dependence on the dimension $d$ from exponential to polynomial.

## 3 CONCLUSION

The focus of this paper is to analyze the error bound of ReLU-activated FNNs in batch-based RL, which are used for approximating value functions in DFQI. Our study provides a clear understanding of how hyper-parameters such as the number of samples, ambient dimension, width and depth of the neural network, and the number of iterations affect the convergence rate. This knowledge can be applied to the training of DFQI and other value-based RL algorithms.

URM STATEMENT

The authors acknowledge that at least one key author of this work meets the URM criteria of ICLR 2023 Tiny Papers Track.

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

## A  BACKGROUND AND NOTATIONS

### A.1  MARKOV DECISION PROCESS

The primary purpose of modeling a Markov decision process (MDP) is to facilitate the derivation of a quintuple $(\mathcal{X}, \mathcal{A}, P, \mathcal{R}, \zeta)$ that represents a discounted MDP. Here, $\mathcal{X}$ and $\mathcal{A}$ are the state and action spaces, respectively. The transition probability kernel $P$ is a measurable function on $\mathcal{X} \times \mathcal{A} \subseteq \mathbb{R}^d$ that defines the probability distribution of the next state given the current state and action. Specially, let $\mathcal{M}(\mathcal{X})$ denote the sets of probability measures on $(\mathcal{X}, \mathcal{B}(\mathcal{X}))$, then $P(\cdot|x, a)$ belongs to $\mathcal{M}(\mathcal{X})$ for each pair $(x, a) \in \mathcal{X} \times \mathcal{A}$, and $P(D|\cdot, \cdot)$ is one measurable function on $\mathcal{X} \times \mathcal{A}$ for every $D \in \mathcal{B}(\mathcal{X})$. $\mathcal{R}(\cdot \mid x, a)$ is the distribution of the immediate reward $R(x, a)$, and $\zeta \in [0, 1)$ is the discount factor. The stochastic policy associated with the action at state $x$ is denoted by $\pi(\cdot|x)$. Given an initial distribution $\nu \in \mathcal{M}(\mathcal{X})$, the batch data $\{Z_i\}_{i=1}^n = \{X_i, A_i, R_i, X_i'\}_{i=1}^n$ with $X_i' = X_{i+1}$ is generated by assuming that $X_1 \sim \nu$ and $A_i \sim \pi(\cdot \mid X_i)$, $R_i \sim \mathcal{R}(\cdot \mid X_i, A_i)$, and $X_i' \sim P(\cdot \mid X_i, A_i)$ for $i = 1, \ldots, n$. The work assumes strict stationarity with $\alpha$-mixing for the MDP $\{Z_i\}_{i=1}^n$ as defined in Definition 2.1. This assumption implies that $Z_i$'s have same distributions.

Let the function $Q^\pi(x, a)$ define the action-value function, which is represented as the expected sum of discounted rewards over an infinite horizon for a given policy $\pi$, starting at state $x$ and taking action $a$ at the first step, that is,

$$Q^\pi(x, a) := \mathbb{E}\left[\sum_{i=1}^\infty \zeta^{i-1} R_i \mid X_1 = x, A_1 = a, \pi\right].$$

Let $\mathcal{T}^\pi$ be the Bellman operator for a given policy $\pi$, that is,

$$\mathcal{T}^\pi Q(x, a) := \mathbb{E}R(x, a) + \zeta P^\pi Q(x, a),$$

with

$$P^\pi Q(x, a) := \int P(dx'|x, a) \pi(da'|x') Q(x', a'),$$

then it has the unique fixed point $Q^\pi$. Under the assumption that $R(x, a)$ ranges between 0 and $R_{\max}$ for all $(x, a) \in \mathcal{X} \times \mathcal{A}$, $Q^\pi$ is constrained to the interval $[0, R_{\max}/(1 - \zeta)]$. Assuming the existence of a policy $\pi^*$ that maximizes $Q^\pi$ and satisfies

$$Q^* := Q^{\pi^*}, \tag{4}$$

we obtain $Q^*$ which satisfies the optimal Bellman equation $Q^* = \mathcal{T}^* Q^*$, where $\mathcal{T}^*$ is the optimal Bellman operator defined as

$$\mathcal{T}^* Q(x, a) = \mathbb{E}[R(x, a)] + \zeta \mathbb{E}_{X' \sim P(\cdot|x, a)} \max_{a' \in \mathcal{A}} [Q(X', a')]. \tag{5}$$

Furthermore, the optimal Bellman operator $\mathcal{T}^*$ is shown to be a $\zeta$-contraction in the sup-norm. The greedy policy $\pi(x; Q)$ for an action-value function $Q$ is defined as the action that maximizes $Q(x, a)$ for a given state $x$, i.e.,

$$\pi(x; Q) \in \operatorname*{argmax}_{a \in \mathcal{A}} Q(x, a), \ x \in \mathcal{X}.$$

## A.2 ReLU-ACTIVATED FEEDFORWARD NEURAL NETWORKS

We now introduce the FNNs with ReLU activations. We use $\mathcal{F}$ to denote the class of FNNs $f_\theta :$ $\mathbb{R}^d \to \mathbb{R}$ with parameter $\theta$, depth $\mathcal{D}$, width $\mathcal{W}$, where $f_\theta$ is defined as $f_\theta(x) = v_\mathcal{D} \circ \rho \circ v_{\mathcal{D}-1} \circ \rho \circ \cdots \circ \rho \circ v_1 \circ \rho \circ v_0(x)$, $x \in \mathbb{R}^d$, and therefore $\|f_\theta\|_\infty \leq \mathcal{B}$ holds for some $0 < \mathcal{B} < \infty$, where $\|\cdot\|_\infty$ refers to the sup-norm, $\rho(x) = \max(0, x)$ is the ReLU activation function operates that pointwisely on $x$ and

$$v_i(x) = \widetilde{A}_i x + b_i, \quad i = 0, 1, \ldots, \mathcal{D},$$

$\widetilde{A}_i \in \mathbb{R}^{d_{i+1} \times d_i}$ is the weight matrix, $b_i \in \mathbb{R}^{d_{i+1}}$ is the bias vector, and $d_i$ is the width of the $i$-th layer. The first layer takes the input data and the last layer gives the output target. The FNNs $f_\theta$ has $\mathcal{D}$ hidden layers and in total $(\mathcal{D}+1)$ layers. We use a $(\mathcal{D}+1)$-vector $(d_0, d_1, \ldots, d_\mathcal{D})^\top$ to describe the width of each layer; in particular, $d_0 = d$ is the dimension of the input $(X, A)$ and $d_\mathcal{D} = 1$ is the dimension of the output. The width $\mathcal{W}$ is defined as the maximum width of hidden layers, i.e., $\mathcal{W} = \max\{d_1, \ldots, d_\mathcal{D}\}$.

## A.3 OTHER NOTATIONS

We introduce other notations used throughout this paper and list them in Table 1.

Table 1: Table of notations used throughout this paper

| Notation | Meaning |
|---|---|
| $a \vee b$ | $\max\{a, b\}, a, b \in \mathbb{R}$. |
| $\lfloor a \rfloor$ | The largest integer less than $a$, $a \in \mathbb{R}$. |
| $\lceil a \rceil$ | The smallest integer no less than $a$, $a \in \mathbb{R}$. |
| $\|x\|_q$ | $\ell_q$-norm of vector $\|x\|_q = (\sum_{i=1}^d |x_i|^q)^{\frac{1}{q}}$, $q \in [1, \infty]$, $x = (x_1, \ldots, x_d)^\top \in \mathbb{R}^d$. |
| $\mathbb{N}_0$ | Non-negative integers. |
| $\mathbb{N}$ | Strictly positive integers. |
| $\|Q\|_{L^q(\mu)}^q$ | $\ell_q$-norm of measurable function $Q : \mathbb{R}^d \to \mathbb{R}^1$ for probability measure $\mu$ on $\mathbb{R}^d$, i.e., $\|Q\|_{L^q(\mu)}^q = \mathbb{E}_{x \sim \mu}|Q(x)|^q$. |

# B DFQI ALGORITHM

The detailed architecture of DFQI is summarized in Algorithm 1.

---

**Algorithm 1** Deep Fitted $Q$-Iteration Algorithm

---

1: Input: Initial value $\widehat{Q}_0 \in \mathcal{F}$.
2: **for** $j = 1, \ldots, J$ **do**
3:     Sampling $(X_i, A_i, R_i, X_i'), i = 1, \ldots, n$.
4:     Compute $Y_i = R_i + \zeta \max_{a' \in \mathcal{A}} \widehat{Q}_{j-1}(X_i', a')$.
5:     Obtain the $j$-step action-value function $\widehat{Q}_j$ via solving (3), that is,

$$\widehat{Q}_j \in \arg\min_{Q \in \mathcal{F}} \widehat{\mathcal{L}}(Q).$$

6: **end for**
7: Output: Estimator $\widehat{Q}_J$ of $Q^*$ and the greed policy $\pi_J = \pi(\cdot; \widehat{Q}_J)$.

---

# C ERROR ANALYSIS

In this section, we provide an error analysis of DFQI by constraining $\|Q^* - Q^{\pi_J}\|_{L_1(\nu)}$ for any permissible distribution $\nu$. We introduce concentration coefficients that regulate the shift in distribution

since specific concentrability is required for the theoretical analysis of RL (Munos, 2003; Xie & Jiang, 2020; 2021; Chen & Jiang, 2019).

**Definition C.1.** *(Concentration Coefficients). Let $\nu_1, \nu_2 \in \mathcal{M}(\mathcal{X} \times \mathcal{A})$ be two probability measures that are absolutely continuous with respect to the Lebesgue measure on $\mathcal{X} \times \mathcal{A}$. Let $\{\pi_t\}_{t \geq 1}$ be a sequence of policies. Suppose the initial state-action pair $(X_0, A_0)$ of the MDP has distribution $\nu_1$, and we take action $A_t$ according to the policy $\pi_t$. For any integer $m$, we denote the distribution of $\{(X_t, A_t)\}_{t=0}^m$ by $\nu_1 P^{\pi_1} P^{\pi_2} \cdots P^{\pi_m}$. The $m$-th concentration coefficient is defined as*

$$c_{\nu_1, \nu_2}(m) = \sup_{\pi_1, \ldots, \pi_m} \left\| \frac{d\left(\nu_1 P^{\pi_1} P^{\pi_2} \ldots P^{\pi_m}\right)}{d\nu_2} \right\|_\infty,$$

*where the supremum is taken over all possible policies. Furthermore, let $\mu$ be the distribution of $(X_i, A_i)$ in Algorithm 1 and let $\nu$ be a fixed distribution on $\mathcal{X} \times \mathcal{A}$. Denote*

$$C_{\nu, \mu} := (1 - \zeta)^2 \cdot \sum_{m \geq 1} m \zeta^{m-1} c_{\nu, \mu}(m), \tag{6}$$

*and assume $C_{\nu, \mu} < \infty$, where $(1 - \zeta)^2$ in (6) is a normalization term, since $\sum_{m > 1} \zeta^{m-1} \cdot m = (1 - \zeta)^{-2}$.*

Next, we introduce the error propagation (Antos et al., 2008; Farahmand et al., 2016; Fan et al., 2020) such that we can relate the error bound of $\|Q^* - Q^{\pi_J}\|_{L_1(\nu)}$ into that of $\|\widehat{Q}_j - \mathcal{T}^* \widehat{Q}_{j-1}\|_{L_2(\mu)}$.

**Proposition C.1.** *(Error propagation) Let $\pi_J$ be the greedy policy of $\widehat{Q}_J$ in Algorithm 1 and $Q^{\pi_J}$ be the action-value function corresponding to $\pi_J$, then*

$$\|Q^* - Q^{\pi_J}\|_{L_1(\nu)} \leq \frac{2\zeta}{(1 - \zeta)^2} \left( C_{\nu, \mu} \max_{1 \leq j \leq J} \|\varepsilon_j\|_{L_2(\mu)} + 2\zeta^J R_{\max} \right),$$

*where $\varepsilon_j = \widehat{Q}_j - \mathcal{T}^* \widehat{Q}_{j-1}$, $j = 1, \ldots, J$.*

Proposition C.1 implies that we only need to bound $\|\widehat{Q}j - \mathcal{T}^* \widehat{Q}_{j-1}\|_{L_2(\mu)}$. To achieve this goal, we start by decomposing the excess risk $\mathcal{L}(\widehat{Q}_j) - \mathcal{L}(\mathcal{T}^* \widehat{Q}_{j-1})$ into two parts, namely the approximation and statistical errors, as shown in Lemma C.1. We then apply suitable techniques from the empirical process with dependent data and deep approximation theory to derive the bound of each of these errors.

**Lemma C.1.** *Given a random sample $\{Z_i\}_{i=1}^n$, the excess risk satisfies*

$$\mathcal{L}(\widehat{Q}_j) - \mathcal{L}(\mathcal{T}^* \widehat{Q}_{j-1}) \leq 2 \sup_{Q \in \mathcal{F}} |\mathcal{L}(Q) - \mathcal{L}(Q)| + \inf_{Q \in \mathcal{F}} \|Q - \mathcal{T}^* \widehat{Q}_{j-1}\|_{L^2(\mu)}^2.$$

## C.1 STATISTICAL ERROR

The statistical error of ReLU-activated FNNs $\mathcal{F}$ with dependent data $\{Z_i\}_{i=1}^n$ is represented by the term $\sup_{Q \in \mathcal{F}} \left| \mathcal{L}(Q) - \widehat{\mathcal{L}}(Q) \right|$. In order to bound this term, we follow the approach described in Modha & Masry (1996) and derive the tail probability bound of the empirical process with $\alpha$-mixing data, indexed by functions in $\mathcal{F}$, using the covering number of $\mathcal{F}$. The covering number can be further bounded by the width and depth of the ReLU-activated neural networks, using VC dimension (Bartlett et al., 2019). Finally, we present the bound on the statistical error, $\sup_{Q \in \mathcal{F}} \left| \mathcal{L}(Q) - \widehat{\mathcal{L}}(Q) \right|$, in the following theorem.

**Theorem C.1.** *Suppose that $\{Z_i\}_{i=1}^n$ is strictly exponentially $\alpha$-mixing, then*

$$\mathbb{E} \sup_{Q \in \mathcal{F}} \left| \mathcal{L}(Q) - \widehat{\mathcal{L}}(Q) \right| \leq C_1 \cdot \left[ \left( \frac{\mathcal{D}^2 \mathcal{W}^2 \log(\mathcal{W}\mathcal{D}) \log(n)}{n^{\frac{\eta}{1+\eta}}} \right)^{1/2} \right],$$

*where $C_1$ is a constant depending on $\mathcal{B}, \eta, a, \bar{\alpha}, R_{\max}$.*

*Proof.* Denote the composite function class

$$\ell \circ \mathcal{F} := \left\{ \ell_Q : \ell_Q(x, a, r, x') = \left( Q(x, a) - r - \gamma \max_{a' \in A} \widehat{Q}_{j-1}(x', a') \right)^2, Q \in \mathcal{F} \right\}.$$

Then, we have

$$\sup_{Q \in \mathcal{F}} \left| \widehat{\mathcal{L}}(Q) - \mathcal{L}(Q) \right| = \sup_{Q \in \mathcal{F}} \left| \frac{1}{n} \sum_{i=1}^{n} (Q(X_i, A_i) - Y_i)^2 - \mathbb{E} (Q(X_i, A_i) - Y_i)^2 \right|$$

$$= \sup_{Q \in \mathcal{F}} \left| \frac{1}{n} \sum_{i=1}^{n} \ell_Q(X_i, A_i, R_i, X'_i) - \mathbb{E}\ell_Q(X_i, A_i, R_i, X'_i) \right|.$$

Let $\text{VC}_{\mathcal{F}}$ be the VC-dimension of $\mathcal{F}$. For any $\delta \geq 0$, we obtain

$$\mathbb{E} \sup_{Q \in \mathcal{F}} \left| \frac{1}{n} \sum_{i=1}^{n} \ell_Q(X_i, A_i, R_i, X'_i) - \mathbb{E}\ell_Q(X_i, A_i, R_i, X'_i) \right|$$

$$\leq \delta + \int_{\delta}^{2\widetilde{M}} P\left( \sup_{Q \in \mathcal{F}} \left| \frac{1}{n} \sum_{i=1}^{n} \ell_Q(X_i, A_i, R_i, X'_i) - \mathbb{E}\ell_Q(X_1, A_1, R_1, X'_1) \right| > \varepsilon \right) d\varepsilon$$

$$\leq \delta + \int_{\delta}^{2\widetilde{M}} 2C\mathcal{N}_n \left( \varepsilon/4, \ell \circ \mathcal{F}, \| \cdot \|_{\infty} \right) \exp\left( -\frac{3n^{(\eta)}\varepsilon^2}{96\widetilde{M}^2 + 32\widetilde{M}\varepsilon} \right) d\varepsilon$$

$$\leq \delta + \int_{\delta}^{2\widetilde{M}} 2C\mathcal{N}_n \left( \frac{\varepsilon}{4\lambda}, \mathcal{F}, \| \cdot \|_{\infty} \right) \exp\left( -\frac{3n^{(\eta)}\varepsilon^2}{96\widetilde{M}^2 + 32\widetilde{M}\varepsilon} \right) d\varepsilon$$

$$\leq \delta + \int_{\delta}^{2\widetilde{M}} 2C \left( \frac{e\mathcal{B}n}{\frac{\delta}{4\lambda} \cdot \text{VC}_{\mathcal{F}}} \right)^{\text{VC}_{\mathcal{F}}} \exp\left( -\frac{3n^{(\eta)}\varepsilon^2}{96\widetilde{M}^2 + 32\widetilde{M}\varepsilon} \right) d\varepsilon$$

$$\leq \delta + 4C\widetilde{M} \left( \frac{4\lambda e\mathcal{B}n}{\delta \text{VC}_{\mathcal{F}}} \right)^{\text{VC}_{\mathcal{F}}} \exp\left( -\frac{3n^{(\eta)}\delta^2}{160\widetilde{M}^2} \right)$$

$$\leq C_1 n^{-\frac{\eta}{2(1+\eta)}} \cdot \sqrt{\log n \cdot \text{VC}_{\mathcal{F}}}.$$

Here, the first inequality holds since $\ell \circ \mathcal{F}$ is bounded by $\widetilde{M} := 6\mathcal{B}^2 + 3R_{\max}^2$, the second inequality holds by some elementary calculations and Theorem 4.3 of Modha & Masry (1996) with $C := 1 + 4e^{-2}\bar{\alpha}$, and $\mathcal{N}_n$ refers to the uniform covering number (Anthony et al., 1999), the third inequality holds since

$$|\ell_{Q_1}(x, a, r, x') - \ell_{Q_2}(x, a, r, x')| \leq \lambda \cdot \|Q_1 - Q_2\|_{\infty}$$

with $\lambda := 4\mathcal{B} + 2R_{\max}$, the fourth inequality holds by the relationship between the covering number and the VC-dimension of the ReLU-activated networks $\mathcal{F}$ (Anthony et al., 1999) given by

$$\mathcal{N}_n \left( \frac{\varepsilon}{4\lambda}, \mathcal{F}, \| \cdot \|_{\infty} \right) \leq \left( \frac{e\mathcal{B}n}{\frac{\varepsilon}{4\lambda} \cdot \text{VC}_{\mathcal{F}}} \right)^{\text{VC}_{\mathcal{F}}},$$

and the last inequality holds for constant $C_1$ depending on $\mathcal{B}, \eta, a, \bar{\alpha}, R_{\max}$ due to the fact that $n^{(\eta)} \geq 2^{-\frac{2\eta+5}{1+\eta}} a^{\frac{1}{1+\eta}} n^{\frac{\eta}{1+\eta}}$ when $\lceil t \rceil \leq 2t$ for all $t \geq 1$ and $\lfloor t \rfloor \geq t/2$ for all $t \geq 2$ and setting

$$\delta^2 = \frac{160\widetilde{M}^2}{n^{\frac{\eta}{1+\eta}}} \text{VC}_{\mathcal{F}} \log\left( \frac{4\lambda e\mathcal{B}n}{\text{VC}_{\mathcal{F}}} \right).$$

Hence, we have

$$\mathbb{E} \sup_{Q \in \mathcal{F}} \left| \widehat{\mathcal{L}}(Q) - \mathcal{L}(Q) \right| \leq C_1 \cdot \left[ \left( \frac{\mathcal{D}^2\mathcal{W}^2 \log(\mathcal{W}\mathcal{D}) \log(n)}{n^{\frac{\eta}{1+\eta}}} \right)^{1/2} \right],$$

where the inequality holds since the upper bound of VC-dimension for the ReLU-activated network $\mathcal{F}$ satisfies

$$c_1 \cdot \mathcal{D}^2\mathcal{W}^2 \log(\mathcal{D}\mathcal{W}^2) \leq \text{VC}_{\mathcal{F}} \leq c_2 \cdot \mathcal{D}^2\mathcal{W}^2 \log(\mathcal{D}^2\mathcal{W}^2)$$

with universal constant $c_1$ and $c_2$, see Bartlett et al. (2019). $\qquad\square$

## C.2 APPROXIMATION ERROR

The term $\inf_{Q \in \mathcal{F}} \|Q - \mathcal{T}^* \widehat{Q}_{j-1}\|^2_{L^2(\mu)}$ can be comtrolled by the approximation error of ReLU-activated FNNs $\mathcal{F}$ to the Hölder class. This is because the smoothing property of $\mathcal{T}^*$ implies that $\mathcal{T}^* \widehat{Q}_{j-1}$ is included in the Hölder class. In order to achieve this, we make the assumption that the distribution of the state-action pair $(X, A)$ is supported on $[0, 1]^d$, without any loss of generality. Furthermore, we need to satisfy the representation condition that the target $Q^*$ is an element of the Hölder class $\mathcal{H}^\gamma$, which is defined as follows.

**Definition C.2.** *(Hölder class) For $\gamma > 0$ with $\gamma = s + r$, where $s \in \mathbb{N}_0$ and $r \in (0, 1]$ and $d \in \mathbb{N}$, we denote Hölder class $\mathcal{H}^\gamma$ as*

$$\mathcal{H}^\gamma = \left\{ f : [0,1]^d \to \mathbb{R}, \max_{\|\widetilde{\alpha}\|_1 \le s} \left\| \partial^{\widetilde{\alpha}} f \right\|_\infty \le B, \max_{\|\widetilde{\alpha}\|_1 = s} \sup_{x \ne y} \frac{|\partial^{\widetilde{\alpha}} f(x) - \partial^{\widetilde{\alpha}} f(y)|}{\|x - y\|_\infty^r} \le B \right\}.$$

We apply the approximation result of Jiao et al. (2023) giving the approximation error bound for Hölder continuous functions using ReLU-activated FNNs, shown in the following Theorem C.2. Note that the prefactor $(d^{\lfloor \gamma \rfloor + (\gamma \vee 1)/2})$ depends on the ambient dimension $d$ polynomially, which improves that in Lu et al. (2021) from exponentially to polynomially.

**Theorem C.2.** *(Theorem 3.3 of Jiao et al. (2023)) Assume that $f \in \mathcal{H}^\gamma$ with $\gamma = s + r, s \in \mathbb{N}_0$ and $r \in (0, 1]$. For any $W, L \in \mathbb{N}$, there exists a function $\tilde{f}$ belonging to the ReLU-activated FNNs $\mathcal{F}$ with width $\mathcal{W} = 38(\lfloor \gamma \rfloor + 1)^2 d^{\lfloor \gamma \rfloor + 1} W \lceil \log_2(8W) \rceil$ and depth $\mathcal{D} = 21(\lfloor \gamma \rfloor + 1)^2 L \lceil \log_2(8L) \rceil$ such that*

$$|f(x) - \tilde{f}(x)| \le 18B(\lfloor \gamma \rfloor + 1)^2 d^{\lfloor \gamma \rfloor + (\gamma \vee 1)/2} (WL)^{-2\gamma/d},$$

*for all $x \in [0,1]^d \backslash \Omega([0,1]^d, S, \delta)$, where $\Omega([0,1]^d, S, \delta) = \cup_{i=1}^d \{x = [x_1, x_2, \ldots, x_d]^\top : x_i \in \cup_{k=1}^{S-1}(k/S - \delta, k/S)\}$, with $S = \lceil (WL)^{2/d} \rceil$ and $\delta \in (0, 1/(3S)]$.*

## C.3 BOUNDING THE EXCESS RISK $\mathcal{L}(\widehat{Q}_j) - \mathcal{L}(\mathcal{T}^* \widehat{Q}_{j-1})$

With Theorems C.1-C.2, we can establish the non-asymptotic error bound for the excess risk $\mathcal{L}(\widehat{Q}_j) - \mathcal{L}(\mathcal{T}^* \widehat{Q}_{j-1}) \left( \|\widehat{Q}_j - \mathcal{T}^* \widehat{Q}_{j-1}\|^2_{L_2(\mu)} \right)$ by choosing appropriate width $\mathcal{W}$ and depth $\mathcal{D}$, shown in Theorem C.3.

**Theorem C.3.** *Suppose that $\{\mathcal{T}^* \widehat{Q}_{j-1}\}_{j=1}^J \in \mathcal{H}^\gamma$ in Definition C.2 with $\gamma = s + r, s \in \mathbb{N}_0$ and $r \in (0, 1]$, $\{Z_i\}_{i=1}^n$ is strictly exponentially $\alpha$-mixing, and the probability distribution $\mu$ of $(X, A)$ is absolutely continuous with respect to Lebesgue measure. Then, for the ReLU-activated FNNs $\mathcal{F}$ with the width $\mathcal{W} = \mathcal{O}\left( (n^{\frac{\eta}{1+\eta}})^{\frac{d}{4(d+4\gamma)}} \log n \right)$ and depth $\mathcal{D} = \mathcal{O}\left( (n^{\frac{\eta}{1+\eta}})^{\frac{d}{4(d+4\gamma)}} \log n \right)$, the excess risk satisfies*

$$\mathbb{E}\left[ \|\widehat{Q}_j - \mathcal{T}^* \widehat{Q}_{j-1}\|^2_{L_2(\mu)} \right] \le C \left[ d^{2s+(\gamma \vee 1)} (n^{\frac{\eta}{1+\eta}})^{\frac{-2\gamma}{d+4\gamma}} (\log n)^3 \right], \quad j = 1, \ldots, J,$$

*where $C$ is a constant depending on $s, B, \mathcal{B}, R_{\max}, \eta, a, \bar{\alpha}$.*

*Proof.* By Theorem C.2, for any $f^* \in \mathcal{H}^\gamma$, there exists one function $\tilde{f} \in \mathcal{F}$ with width $\mathcal{W} = 38(s+1)^2 d^{s+1} W \lceil \log_2 8W \rceil$ and depth $\mathcal{D} = 21(s+1)^2 L \lceil \log_2 8L \rceil$ such that

$$\left| f^*(x) - \tilde{f}(x) \right| \le 18B(s+1)^2 d^{s+(\gamma \vee 1)/2} (WL)^{-2\gamma/d}$$

for $x \in \cup_\theta \widetilde{Q}_\theta$, with

$$\widetilde{Q}_\theta = \left\{ x : x_i \in \left[ \frac{\theta_i}{S}, \frac{\theta_i + 1}{S} - \delta \cdot 1_{\{\theta_i < S-1\}} \right] \right\}$$

where $\theta = (\theta_1, \theta_2, \ldots, \theta_n) \in \{0, 1, \ldots, S-1\}^d$, and $\delta$ is an arbitrary number satisfying $0 < \delta \le \frac{1}{3S}$. Then the Lebesgue measure of $[0,1] \backslash \widetilde{Q}_\theta$ is no more than $dS\delta$ which can be arbitrarily small if $\delta$ is arbitrarily small. Since $\mu$ is absolutely continuous with respect to the Lebesgue measure and $\mathcal{T}^* \widehat{Q}_{j-1} \in \mathcal{H}^\gamma$, we have

$$\inf_{Q \in \mathcal{F}} \|Q - \mathcal{T}^* \widehat{Q}_{j-1}\|^2_{L^2(\mu)} \le 324B^2(s+1)^4 d^{2s+(\gamma \vee 1)} \left\lfloor (WL)^{2/d} \right\rfloor^{-2\gamma}.$$

By Lemma C.1 and Theorem C.1, it yields that

$$\mathbb{E}[\|\widehat{Q}_j - \mathcal{T}^*\widehat{Q}_{j-1}\|^2_{L_2(\mu)}] \leq C_1 \cdot \left[ \left( \frac{\mathcal{W}^2\mathcal{D}^2 \log(\mathcal{W}\mathcal{D}) \log(n)}{n^{\frac{\eta}{1+\eta}}} \right)^{1/2} \right.$$
$$\left. + 324B^2(s+1)^4 d^{2s+(\gamma \vee 1)} \left\lfloor (WL)^{2/d} \right\rfloor^{-2\gamma} \right].$$

Setting $\mathcal{W} = \mathcal{O}\left( (n^{\frac{\eta}{1+\eta}})^{\frac{d}{4(d+4\gamma)}} \log n \right)$ and depth $\mathcal{D} = \mathcal{O}\left( (n^{\frac{\eta}{1+\eta}})^{\frac{d}{4(d+4\gamma)}} \log n \right)$, then it yields that

$$\mathbb{E}[\|\widehat{Q}_j - \mathcal{T}^*\widehat{Q}_{j-1}\|^2_{L_2(\mu)}] \leq C_2 \cdot \left[ d^{2s+(\gamma \vee 1)} (n^{\frac{\eta}{1+\eta}})^{\frac{-2\gamma}{d+4\gamma}} (\log n)^3 \right],$$

where $C_2$ is a constant depending on $s, B, \mathcal{B}, R_{\max}, \eta, a, \bar{\alpha}$.

$\square$

### C.4 PROOF OF THEOREM 2.1

*Proof.* By Proposition C.1 and Theorem C.3, we can conclude that

$$\mathbb{E}\left[ \|Q^* - Q^{\pi_J}\|_{L_1(\nu)} \right] \leq \frac{2\sqrt{C}C_{\nu,\mu}\zeta}{(1-\zeta)^2} \cdot \left[ d^{s+(\gamma \vee 1)/2} (n^{\frac{\eta}{1+\eta}})^{\frac{-\gamma}{d+4\gamma}} (\log n)^{3/2} \right] + \frac{4\zeta^{J+1}}{(1-\zeta)^2} R_{\max},$$

where $C$ is a constant depending on $s, B, \mathcal{B}, R_{\max}, \eta, a, \bar{\alpha}$. This completes the proof. $\square$

