# OpenReview forum: "Error Analysis of Fitted Q-iteration with ReLU-activated Deep Neural Networks"
_ICLR.cc/2023/TinyPapers — Submitted to Tiny Papers @ ICLR 2023_

### Official Review · Reviewer_ssyR · 2023-03-27

**Confidence:** 3

**Summary Of Contributions:**

This paper studies the error analysis of the deep-fitted Q-iteration (DFQI) with value functions approximation using ReLU networks in batch RL. The bound depends on the sample complexity, the ambient dimension polynomially, the width and depth of the neural network.

**Rating:**

High Potential (HP): a submission which meets the reviewing criteria and has potential to make an impact on the field

**Strengths And Weaknesses:**

Strengths
- The paper is generally well-written.
- The theoretical results improved upon existing work of Fan et al. (2020) by introducing a weaker $\alpha-$mixing condition; the error bound is also improved by considering the sample complexity and reducing the dependency on the dimension d from exponential to polynomial. Overall, I consider this result to be strong.

Weakness
- It would be helpful if the authors include a Table to summarize the notations used in this work.
- The $\alpha-$mixing condition should be stated in the main paper.

**Suggested Changes:**

Please see Weakness

---

> ### Author Response · Authors · 2023-04-18
> **Reply to Reviewer ssyR**
>
> Thank you for the detailed comment. We have revised accordingly in the updated version.

---

### Official Review · Reviewer_maBM · 2023-03-30

**Confidence:** 2

**Summary Of Contributions:**

This paper provides an error analysis of deep-fitted Q-iteration (DFQI) applying ReLU-activated feed-forward neural networks in reinforcement learning. This analysis seeks to improve a previous study by considering more relevant variables such as the ambient dimension, the number of samples, the width and depth of the neural network, and the number of training iterations.

**Rating:**

Clear, Correct, and Reproducible (CCR): a submission which meets the reviewing criteria

**Strengths And Weaknesses:**

*Strengths:
- The paper is well-written and structured
- The paper provides insights into hyperparameters setting when training DFQI to achieve a desired convergence rate.
- The authors analyzed the error with detailed proofs step by step (statistical error, approximation error, etc).

*Weaknesses:
- The study doesn't describe if there are limitations and what would be the future work.

**Suggested Changes:**

*Suggested Changes
- I recommend briefly explaining why ReLU was chosen and not other activation function.
- I suggest briefly highlighting the impact of this study in the abstract.

---

> ### Author Response · Authors · 2023-04-18
> **Reply to Reviewer maBM**
>
> Thanks for the comment. The main contributions of the paper are as follows: A non-asymptotic error bound, through the tools of error propagation, empirical process, and deep approximation, has been established between the estimated action-value function corresponding to the estimated greedy policy and the optimal  $Q^*$ by controlling the statistical and approximation errors on the Markov decision process data assumed to be $\alpha$-mixing. We present the first generalization bound in terms of the ReLU-activated FNN  and $\alpha$-mixing data in the context of reinforcement learning. We demonstrate that the error bound depends on the sample size, the ambient dimension (polynomially), the width and depth of the neural network, and the number of training iterations, and thus provide a powerful tool in future hyper-parameters setting of deep neural networks in DFQI.

---

### Comment · Area_Chair_wAzw · 2023-06-02
**ICLR Tiny Paper Archival**

This work meets the threshold for archival, contents the URM statement and is deanonymized.

---

### Meta-Review · Area_Chair_wAzw · 2023-04-09

**Recommendation:** Invite to present
**Confidence:** 4

**Metareview:**

This paper studies error analysis of deep-fitted Q-iteration (DFQI) in reinforcement learning, by considering more relevant variables than the state-of-the-art such as the ambient dimension, the number of samples, the width and depth of the neural network, and the number of training iterations. **This paper is very well-written and the conclusions drawn are substantiated by formal analysis**. The resulting findings can help hyperparameters selection for DFQI or other RL algorithms to achieve a desired convergence rate.

**Summary:**

The paper studies the error analysis of deep-fitted Q-iteration (DFQI) with value function approximation using ReLU networks in Reinforcement Learning. Reviewers suggested including a discussion on limitations and future work. Reviewers also recommended motivating the ReLU choice.

**Reason For Not Giving A Higher Recommendation:**

Error bounds are specifically given for the ReLU activation function.


**Reason For Not Giving A Lower Recommendation:**

The theoretical results improved upon the existing work of Fan et al. (2020) and promise to be impactful for hyper-parameters selection when training DFQI.

---

> ### Author Response · Authors · 2023-04-18
> **Reply to Area Chair wAzw**
>
> Thank you for the comments. We have updated the manuscript according to the suggestions from reviewers.

---

### Decision · Program_Chairs · 2023-04-09

Invite to present